The influence of rhizosphere microbial diversity on the accumulation of active compounds in farmed Scutellaria baicalensis

Dong Ping 1
Lu Yang 2
Bai Jia 1
Zhao Chunying 1 zhaochunying1908@163.com
Li Zhongsi 1 lzs8680@126.com
Cao Yu 1
Li Yingzhe 1
1 Institute of Traditional Chinese Medicine, Chengde Medical College , Chengde City, Hebei Province , China
2 Sericulture Research Institute, Chengde Medical College , Chengde City, Hebei Province , China
De Costa Janendra
Electronic publication date: 2024 Dec 24
Publication date: 2024
Volume: 12
Electronic Location ID: e18749
Received 2024 Sep 11; Accepted 2024 Dec 2
Copyright: © 2024 Dong et al.
Copyright year: 2024
Copyright holder: Dong et al.
License: This is an open access article distributed under the terms of the Creative Commons Attribution License, which permits unrestricted use, distribution, reproduction and adaptation in any medium and for any purpose provided that it is properly attributed. For attribution, the original author(s), title, publication source (PeerJ) and either DOI or URL of the article must be cited.
License URL: https://creativecommons.org/licenses/by/4.0/

Keywords: Scutellaria baicalensis, Bacteria, Fungi, Active compounds, Quality

Funding: Natural Science Foundation Program H2022406079 Technical Research Program QN2022082 This work was supported by the Specialized Master of Traditional Chinese Medicine, Natural Science Foundation Program of Hebei Province (H2022406079) and the Technical Research Program for Higher Education Institutions in Hebei Province (QN2022082). The funders had no role in study design, data collection and analysis, decision to publish, or preparation of the manuscript.

==============================
Rhizosphere microorganisms are important factors affecting herb quality and secondary metabolite accumulation. In this study, we investigated the diversity of rhizosphere microbial communities (bacteria and fungi) and their correlations with soil physicochemical properties and active compounds of Scutellaria baicalensis (baicalin, oroxindin, baicalein, wogonin, and oroxylin A) from cultivated Scutellaria baicalensis with three different origins via high-throughput sequencing and correlation analysis to further clarify the role of soil factors in the accumulation of the active compounds of Scutellaria baicalensis. The results are summarized as follows. A total of 28 dominant bacterial genera, such as Arthrobacter, Rubrobacter, Microvirga, and Sphingomonas, and 42 dominant fungal genera, such as Alternaria, Spegazzinia, and Minimedusa, were detected. The soil microbial communities associated with cultivated Scutellaria baicalensis were very diverse, but there were some differences in the relative abundances of microbial taxa. Correlation analysis revealed that the bacterial genera Rubrobacter, Ellin6055, Gaiella, norank__f__norank__o___norank__c__bacteriap25, unclassified__f__Micromonosporaceae, norank__f__ Gemmatimonadaceae, Arthrobacter, and Sphingomonas and the fungal genera Tausonia, Minimedusa, Cercospora, Botrytis, Alternaria, Boeremia, Titaea, Solicoccozyma, and Mortierella were positively or negatively correlated with each active component of Scutellaria baicalensis and were important genera affecting the accumulation of the active compounds of Scutellaria baicalensis and correlated with soil physiochemistry to different degrees. These results suggest that rhizosphere microorganisms may play a role in the accumulation of active compounds in medicinal plants and provide a scientific basis for guiding the cultivation of Scutellaria baicalensis, developing biofertilizers, and improving the quality of Scutellaria baicalensis medicinal materials.

Introduction

Scutellaria baicalensis, the dried root of the Labiatae family plant Scutellaria baicalensis Georgi, is effective at alleviating heat, eliminating dampness, detoxifying and preventing diarrhoea, halting bleeding, and calming foetuses (National Pharmacopoeia Commission, 2020). The primary chemical components of Scutellaria baicalensis are flavonoids, which underpin the pharmacological effects of the species. The four most significant flavonoid compounds found in Scutellaria baicalensis are baicalin, baicalein, oroxindin, and wogonin (Karimov & Botirov, 2017; Bai et al., 2020; Song et al., 2021; Qiao et al., 2016). Benzyl alcohol, benzoic acid, β-sitosterol, polysaccharides, terpenoids, and volatile oils are also present in Scutellaria baicalensis in addition to flavonoids (Zhao et al., 2019). Antioxidant, anticancer, anti-inflammatory, antibacterial, cardioprotective, hepatoprotective, nephroprotective, and neuroprotective qualities are all present in Scutellaria baicalensis (Ganguly, Gupta & Pandey, 2022).

The two groups of Scutellaria baicalensis that compose the growth environment are those that are farmed and those that are wild. The supply of Scutellaria baicalensis has steadily changed from wild to cultivated due to the growing market demand for real Chinese herbs and the depletion of wild resources. The uneven quality of the herb lowers the safety and efficacy of the drug’s therapeutic use. Variations in elements such as the superiority of the seed source and planting management procedures of growing Scutellaria baicalensis have led to this (Wen et al., 2023). Soil factors have been investigated to examine the relationships among soil microorganisms, the most active compounds in the soil, and the quality of herbs, as these factors significantly impact the quality of the plant (Li et al., 2023).

Microorganisms in soil include bacteria, fungi, protozoa, algae, etc., which are not only involved in nutrient cycling and organic matter transformation but also modify soil habitats through a variety of biochemical and biophysical mechanisms (Jansson & Hofmockel, 2020; Zhao et al., 2024; Yang et al., 2022). These organisms change the soil environment via a range of biochemical and biophysical processes, in addition to participating in the cycling of nutrients and the transformation of organic matter (Philippot et al., 2024). Their diversity serves as a sensitive gauge of the quality of the soil, picking up minute alterations in the soil and offering data for assessing soil function (Velmurugan et al., 2022). Rhizosphere soil microorganisms, also referred to as the second genome of plants, can speed up the conversion and storage of effective nutrients in rhizosphere soil, encourage nutrient uptake by the plant root system, and control soil-borne diseases caused by medicinal plants, all of which affect herb quality (Yang et al., 2016; Berendsen, Pieterse & Bakker, 2012). The rhizosphere is an active area of material and energy exchange between plants and soil. The main medicinal component of Scutellaria baicalensis is the root. Studies on soil microorganisms in the rhizosphere zone of Scutellaria baicalensis exist, but these studies focus only on comparing changes in its microbial community under different treatments. There are fewer studies comparing the correlation between soil microorganisms and active compounds of Scutellaria baicalensis from different origins. Research on the relationships between rhizosphere soil microorganisms and the quality of herbs derived from Scutellaria baicalensis has further elucidated the synergistic relationships among plants, soil, and microorganisms in soil ecosystems. This study analysed the correlation between soil factors and the accumulation of active compounds of Scutellaria baicalensis to provide theoretical support for the artificial cultivation of Scutellaria baicalensis and the standardization of planting bases, as well as to provide a scientific basis for the cultivation of Scutellaria baicalensis varieties with unique advantages that are suitable for production in specific regions (Song et al., 2023). Nevertheless, the intrinsic mechanism by which soil microorganisms in the rhizosphere of Scutellaria baicalensis specifically affect the accumulation of active ingredients is not completely clear and needs to be studied further.

Materials and Methods

Sample collection and processing

The three Scutellaria baicalensis terrestris growing regions around Chengde city in Hebei Province, namely, KC (Kuancheng County), PQ (Pingquan city), and FN (Fengning County), constituted the sites of the experimental sample collection, which occurred between March and April 2023. Figure 1 shows the distribution of sample collection sites. A five-point random sampling method was used to select healthy and disease-free Scutellaria baicalensis herbs. When sampling, the dead leaves on the soil surface were first removed 10 cm away from the Scutellaria baicalensis root system. The Scutellaria baicalensis root system was excavated into a cylindrical soil column with a shovel, the Scutellaria baicalensis herbs were removed, five plants were randomly selected within each sample square, and the intact roots were excavated and mixed to make a one-portion sample of the herbs. The rhizosphere soil was collected via the root-shaking method by removing the roots and gently shaking off the lumps of soil, and the rhizosphere soil that was tightly adhering to the surface of the herbs was collected with a sterile brush; three replicates of equal mixtures were set up (Liu et al., 2024b).

Figure 1 Geographic distribution of sample collection.

Detection of the contents of the active compounds of Scutellaria baicalensis

A one-measurement-multiple assessment approach adopted from Wei (2022) was used to simultaneously detect the active ingredients of Scutellaria baicalensis, viz., baicalin, oroxidin, baicalein, wogon and oroxylin A.

Determination of soil physical and chemical properties

The physical and chemical properties of the soil were determined according to the methods of Bao (2000). Total nitrogen (TN) was determined via the Kjeldahl method; total phosphorus (TP) and effective phosphorus (AP) were determined via the molybdenum antimony colorimetric method; total potassium (TK) was determined via the sodium hydroxide fusion-flame photometer method; soil organic matter (OM) was determined via the potassium dichromate-sulfuric acid dilution preheating method; hydrolysable nitrogen (HN) was determined via the semimicrovolume Kjeldahl method and diffusion method; quick-acting potassium (AK) was determined via the flame photometer method; effective iron (AFe), effective manganese (AMn), effective copper (ACu), and effective zinc (AZn) were determined via the diethylenetriaminepentaacetic acid (DTPA) leaching method; exchangeable calcium (ECa) and exchangeable magnesium (EMg) were determined via the shaking method; and pH was determined via the potentiometric method.

Soil microbial DNA extraction and high-throughput sequencing

An E.Z.N.A.® Soil Kit (Omega Bio-Tek, Norcross, GA, USA) was used to extract total microbial DNA from the soil, and agarose gel electrophoresis with a 1% mass fraction was used to assess the concentration and purity of the extracted DNA. The V3 to V4 sections of the bacterial 16S rDNA gene were amplified via the universal primers 338F and 806R, whereas the fungal ITS1 region was amplified via the primers ITS1F and ITS2R. Four microlitres of 5×FastPfu Buffer, two μL of 2.5 mM dNTPs, 0.8 μL of forward primer (5 μM), 0.8 μL of reverse primer (5 μM), 0.4 μL of FastPfu Polymerase, 0.2 μL of bovine serum albumin (BSA), and 10 ng of template composed the 20 μL amplification system. BSA, 10 ng of template DNA, and 20 μL of ddH2O were added. The following conditions were used for amplification: 3 min at 95 °C for predenaturation, 30 s at 95 °C, 30 s at 55 °C, 45 s at 72 °C, 27 cycles, 10 min at 72 °C for extension, and 10 °C until the reaction was finished. For every sample, three replicates were run. The PCR amplification products from the replicates were combined, found, recovered, and purified, and their fluorescence was measured to create a MiSeq library that was then sequenced both ways.

Data processing

To produce high-quality sequences, the sequencing results were filtered and spliced via the Flash (Version 1.2.11) program (Magoč & Salzberg, 2011). Using Uparse (Version 11) software, the sequences were grouped with a 97% sequence similarity threshold, and chimaeras were eliminated in the process to produce representative sequences of operational taxonomic units (Haas et al., 2011). Mothur (Version 1.30.2) was used to generate the alpha diversity index, which measures the diversity and richness of the microbial community (Alessandri et al., 2019). The beta diversity distance matrix was computed via QIIME (Zhang et al., 2019) (Version 1.9.1), and PCoA (Lutsiv et al., 2021) and charting to examine the similarities or differences between the sample communities were performed via the R language (Version 3.3.1) vegan software package. Using the pheatmap package of R software, correlation heatmap analysis was carried out (Version 3.3.1). With the LDA threshold set at 3.5, linear discriminant analysis was used to assess the impact of soil microbial abundance on the effect of differences, investigate biomarkers, and identify significantly different species in different groups. For multigroup comparisons (differing in multiple groups), a more stringent all-against-all strategy was selected. PICRUSt2 (Version 2.2.0) software was used to examine the metabolic pathways and functions of the soil bacteria, whereas FUNGuild (Version 1.0) was employed to predict the functions of the Scutellaria baicalensis soil fungi (Wang, 2022). Using SPSS Statistics 25.0, one-way analysis of variance (ANOVA, p < 0.05) and general linear bivariate analysis were used to examine the variability among the five active compounds in Scutellaria baicalensis in the three sample plots. The relationships between soil physiochemistry and the five active compounds in the growing Scutellaria baicalensis were examined via Pearson’s linear correlation analysis, with p < 0.05 denoting statistically significant differences. Spearman’s correlation analysis was used to study the correlations between the five active compounds of Scutellaria baicalensis and soil microorganisms and soil physiochemistry.

Results and analysis

Analysis of the contents of the active compounds of Scutellaria baicalensis

As shown in Fig. 2, in the three sample plots, PQ had the highest baicalin content, and KC had the lowest baicalin content, which were significantly different (p < 0.05). PQ had the highest content of oroxindin and wogonin. FN had the lowest content of oroxindin and wogonin. KC had the highest baicalein content and oroxylin A content. FN had the lowest baicalein content and oroxylin A content. The results revealed that there was variability in the active compounds of Scutellaria baicalensis from different origins.

Figure 2 Histograms of the contents of the five active compounds of Scutellaria baicalensis.

(A) Baicalin, (B) oroxindin, (C) baicalein, (D) wogonin, (E) oroxylin A. Different lowercase letters for the same indicator represent significant differences (P < 0.05), and the same lowercase letters represent non-significant differences (P > 0.05).

Rhizosphere soil physicochemical characteristics

Table 1 indicates that there was variability in the physiochemistry of the rhizosphere soils of Scutellaria baicalensis from different origins. Specifically, the contents of TN, TP, OM, AP, AFe, ACu, AZn, and EMg in the three sample plots were significantly different (p < 0.05), with KC being significantly greater than PQ and FN. Conversely, the EC content and PH of FN were significantly greater than those of KC and PQ (p < 0.05), and the TK and AK contents of PQ were significantly greater than those of KC and FN (p < 0.05).

Table 1 Physical and chemical characteristics of the Scutellaria baicalensis rhizosphere soil.

Norm	KC	PQ	FN	
TN/(g/kg)	1.11 ± 0.02a	0.93 ± 0.01b	0.83 ± 0.01c	
TP/(g/kg)	0.96 ± 0.01a	0.58 ± 0.01c	0.70 ± 0.01b	
TK/(g/kg)	19.46 ± 0.10b	20.89 ± 0.10a	19.08 ± 0.10c	
OM/(g/kg)	19.80 ± 0.20a	16.30 ± 0.26b	12.33 ± 0.21c	
HN/(mg/kg)	78.53 ± 1.42a	68.27 ± 1.05b	67.67 ± 1.07b	
AP/(mg/kg)	41.33 ± 0.61a	13.27 ± 0.15b	3.47 ± 0.06c	
AK/(mg/kg)	122.67 ± 2.52c	207.00 ± 2.00a	141.33 ± 1.53b	
AFe/(mg/kg)	11.57 ± 0.15a	9.33 ± 0.25b	8.17 ± 0.21c	
AMn/(mg/kg)	8.50 ± 0.10a	7.80 ± 0.20b	7.77 ± 0.21b	
ACu/(mg/kg)	1.62 ± 0.03a	1.14 ± 0.03b	0.55 ± 0.01c	
AZn/(mg/kg)	2.27 ± 0.04a	0.79 ± 0.01c	1.11 ± 0.03b	
ECa/(g/kg)	3.99 ± 0.03c	6.36 ± 0.05b	8.77 ± 0.05a	
EMg/(g/kg)	0.38 ± 0.01a	0.34 ± 0.01b	0.17 ± 0.01c	
PH	7.41 ± 0.04c	7.80 ± 0.03b	7.90 ± 0.06a	
Note:

The data are presented as the means ± standard deviations (n = 3). While the same lowercase letters indicate no significant differences (p > 0.05), distinct lowercase letters indicate changes in the physical and chemical properties of soils of different origins (p < 0.05).

Correlation analysis of the active compounds of Scutellaria baicalensis and soil physicochemical properties

The correlation analysis results (Table 2) revealed that baicalein, wogonin, and oroxylin A were highly significantly correlated with TP, AMn, and AZn (p < 0.01); baicalin and both TP and AZn had highly significant negative correlations (p < 0.01); oroxindin and TK and EMg had highly significant positive correlations (p < 0.01); and ACu had significant positive correlations (p < 0.05). There were varying degrees of strong correlations between the active compounds of Scutellaria baicalensis and soil physicochemical properties.

Table 2 Correlation analysis of the active compounds of Scutellaria baicalensis and soil physicochemical properties.

	baicalin	oroxindin	baicalein	wogonin	oroxylin A.	
TN	−0.365	0.536	0.982**	−0.566	0.931**	
TP	−0.872**	−0.091	0.873**	−0.954**	0.944**	
TK	0.903**	0.798**	−0.142	0.790*	−0.311	
OM	−0.18	0.692*	0.929**	−0.387	0.851**	
HN	−0.624	0.281	0.972**	−0.769*	0.984**	
AP	−0.47	0.443	0.997**	−0.651	0.968**	
AK	0.994**	0.534	−0.524	0.958**	−0.666	
AFe	−0.378	0.545	0.982**	−0.56	0.938**	
AMn	−0.574	0.316	0.917**	−0.687**	0.925**	
ACu	−0.512	0.710*	0.919**	−0.362	0.836**	
AZn	−0.809**	0.028	0.924**	−0.909**	0.977**	
ECa	0.213	−0.662	−0.941**	0.419	−0.867**	
EMg	0.151	0.869**	0.759**	−0.07	0.63	
PH	0.516	−0.408	−0.988**	0.680*	−0.972**	
Notes:

* Indicates a significant correlation at the 0.05 level (two-tailed).

** Indicates a significant correlation at the 0.01 level (double-tailed).

Analysis of soil microbial diversity

Venn diagram analysis at the OTU level

After valid sequences from every sample were grouped into operational taxonomic units (OTUs) with 97% concordance, species annotations were added to the representative sequences of each OTU. A total of 6,332 bacterial OTUs and 1,453 fungal OTUs were found in KC, 5,782 bacterial OTUs and 1,161 fungal OTUs were found in FN, 6,135 bacterial OTUs and 1,443 fungal OTUs were found in FN, and 6,135 bacterial OTUs and 1,161 fungal OTUs were found in PQ, according to a Venn diagram analysis. A total of 2,609 bacterial OTUs were found in the three sample locations, accounting for 24.25% of all the OTUs (Fig. 3A). With 2,127 bacterial OTUs that were endemic to the site and composed 19.77% of all the OTUs, KC had the most of any location. There were 1,988 and 1,965 bacterial OTUs that were endemic to PQ and FN, respectively, accounting for 18.48 and 16.40% of all the OTUs. As shown in Fig. 3B, the total number of fungal OTUs in the rhizosphere soils of the three samples was 397, or 14.58% of all the OTUs. With a total of 665 OTUs, or 24.43% of all the OTUs, the PQ-specific fungi had the highest number of OTUs. With 460 OTUs, or 16.90% of all the OTUs, the FN-specific fungi had the fewest number of OTUs overall.

Figure 3 Venn diagram of bacterial and fungal OTUs.

(A) Bacterium, (B) fungus.

Alpha diversity analysis

The Shannon index reflects the uniformity of distribution among various individuals; the higher the value is, the greater the diversity of the flora. The Chao index is frequently used to estimate the richness of microorganisms in samples; the higher the index is, the richer the flora. Alpha diversity can reflect the richness and diversity of microbial communities. Using the Kruskal‒Wallis rank sum test (0.95 level of significance), the Shannon index and Chao index of rhizosphere microorganisms in the three sample plots were used as variables to further test whether there was a significant difference in the values of the diversity indices among multiple groups. The p value of the bacterial Shannon index in the three sample plots was 0.02732, which was less than 0.05, indicating that there was a difference in the bacterial Shannon index among the three subgroups. There was a highly significant difference between the bacterial Shannon indices of KC and FN, KC and PQ, and FN and PQ (p < 0.001) (Fig. 4A). There was a small difference in the bacterial Chao index among the three subgroups (Fig. 4B). The fungal Shannon index and Chao index also showed small differences among the three sample sites (p > 0.05) (Figs. 5A and 5B). The above results indicated that there were highly significant differences in the diversity of bacterial communities in the rhizosphere soils of Scutellaria baicalensis from the three sample sites, but the differences in abundance were small, whereas there were small differences in the diversity and abundance of fungi.

Figure 4 Analysis of intergroup differences in bacterial alpha diversity indices.

Note: The presence of an asterisk indicates the presence of variability. ***Significant at the 0.001 probability level.

Figure 5 Analysis of intergroup differences in fungal alpha diversity indices.

Note: The absence of an asterisk indicates that the difference is not significant.

Species composition

Going from genus-level studies to species composition studies reveals the compositional structure of soil microbial communities in more detail and helps in understanding the relationships between microbial diversity and ecological functions. After the genus-level community compositions of the microorganisms in the three rhizosphere soil samples were analysed, those with an abundance of less than 1% were combined with other species, and those with an abundance of more than 1% were classified as the dominant bacterial genera. In the three sample plots, 28 dominant bacterial genera were found, accounting for approximately 60% of the total abundance (Fig. 6A). S1 shows the differences in the relative abundances of several major bacterial species in the various sample plots. Among the three sample plots, norank_f__Vicinamibacteraceae was the bacterial genus that was absolutely dominant. Four genera, Arthrobacter, Sphingomonas, Alternaria, and Mortierella, are important genera in terms of species composition, but there were obvious differences in relative abundance among the three origins. Therefore, analysis of variance (ANOVA) was used to scientifically compare the rhizosphere soil microbial species richness of Scutellaria baicalensis from different origins and to determine whether these differences were caused by various factors of origin. Arthrobacter and Sphingomonas were substantially more prevalent in PQ than in FN and KC (p < 0.01), according to the ANOVA results (Figs. 7A and 7B).

Figure 6 Community composition of the rhizosphere soil of Scutellaria baicalensis at the genus level.

(A) Bacterium, (B) fungus.

Figure 7 Relative abundances of Arthrobacter (A), Sphingospora (B), Alternaria (C) and Mortierella (D) in the three plots.

Note: The presence of an asterisk indicates the presence of variability. p < 0.05 indicates that the index is significantly different between groups; * p < 0.05, *** p < 0.001.

At the three sample sites, 42 fungal species were found to be dominant, accounting for approximately 70% of the overall fungal abundance (Fig. 6B). S2 displays the relative abundance of the prominent fungal genera at each sample site. At sample site FN, Metarhizium had the highest relative abundance of any fungal genus at 26.81%, which was significantly greater than the relative abundances at the other two sample sites. Alternaria presented the highest abundance in PQ (p < 0.001), according to the ANOVA results (Fig. 7C), but the abundance of Mortierella in KC was substantially greater than that in PQ and FN (p < 0.05) (Fig. 7D).

Species variation analysis

A total of 31 bacterial bioindicator species (Fig. 8A) and 34 fungal bioindicator species (Fig. 8B) were detected. These microbial taxa were as follows: among the bacterial taxa, KC enriched a total of four taxa with one class, one order, one family and one genus; PC enriched a total of 10 taxa with one phylum, one class, two orders, four families and one genus; and FN enriched a total of 17 taxa with two phyla, four classes, five orders, three families and three genera. Among the fungal taxa, KC enriched 11 taxa with one class, two orders, four families and four genera; PC enriched 13 taxa with one order, four families and eight genera; and FN enriched 10 taxa with one phylum, one class, two orders, three families and three genera. FN enriched the most differentiated species of bacteria, and PC enriched the most differentiated species of fungi. Species with significant differences in bacteria were enriched in Arthrobacter, Sphingomonas, and norank-Gemmatimonadaceae at the genus level. Among the fungi, those with significant differences were enriched in Alternaria, Boeremia, Botrytis, Cercospora, and Dioszegia at the genus level.

Figure 8 Multilevel species difference judgement analysis via LEfSe.

(A) Bacterium, (B) fungus.

Beta diversity analysis

Principal coordinate analysis (PCoA) is a dimensionality reduction technique that evaluates, in percentage terms, how well each axis accounts for the overall variations in colony structure. It is based on a distance matrix. The higher the similarity between samples is, the more clustered they are; conversely, the lower the similarity between samples is, the higher the degree of dispersion. With a total explanation of 62.32%, Fig. 9A illustrates how the first and second principal components explained 34.24% and 28.08%, respectively, of the variation in the rhizosphere soil bacterial community structure of the three sample plots. With a total explanation of 69.22%, the first and second main components explained 39.57% and 29.65%, respectively, of the variation in fungal community structure (Fig. 9B). The figure shows that the three sample sites were clustered in their respective areas within the bacterial and fungal communities, with PQ distributed on the right side of PC and FN and KC on the left. This suggests that there were variations in the community composition among the individual sample sites as well as similarities in the community composition within each group.

Figure 9 PCoA.

(A) Bacterium, (B) fungus.

Correlations between the active compounds of Scutellaria baicalensis and soil microorganisms

Spearman’s correlation heatmap was used to analyse the relationships between the active compounds of Scutellaria baicalensis and bacterial/fungal genera. Figure 10A shows that baicalein and oroxylin A were significantly negatively correlated with Gaiella and norank_f__Gemmatimonadaceae and highly significantly positively correlated with Marmoricola. The Oroxindin content exhibited a highly significant negative correlation with norank_f__norank_o___0319-7L14 and a highly significant positive correlation with Arthrobacter and Sphingomonas. Oroxylin A and baicalin were positively correlated with certain bacterial genera. This included a positive correlation and varying degrees of significance with Rubrobacter, Ellin6055, unclassified_f__Micromonosporaceae, and norank_f__norank__o_Gaiellales; there was also a negative correlation with MND1, norank__ f__norank__o_norank__c__bacteriap25, and norank__f__norank__o_Vicinamibacterale.

Figure 10 Correlation analysis between the active compounds of Scutellaria baicalensis and the dominant species of soil microorganisms.

(A) Bacterium, (B) fungus.

Figure 10B shows that baicalein and oroxylin A were highly significantly positively correlated with the fungal genera Tausonia and Minimedusa. Oroxindin had highly significant positive correlations with the fungal genera Cercospora, Alternaria, and Botrytis. Baicalin and wogonin had significant positive correlations with unclassified_o___Helotiales, Setophaeosphaeria, Boeremia, unclassified_f__Phaeosphaeriaceae, and Vishniacozyma and with the fungal genera unclassified_o___Helotiales, Setophaeosphaeria, Boeremia, unclassified_f__Phaeosphaeriaceae, Vishniacozyma, Setophaeosphaeria, and Filobasidium; they had significant negative correlation with Titaea, Solicoccozyma, and Chaetomium; and there was a significant negative correlation between the content of wogonin and the fungal genus Mortierella. The results indicated that the soil microorganisms responded significantly to each active compound of Scutellaria baicalensis.

Analysis of the correlations between soil microorganisms and physical and chemical parameters

Figure 11A shows that the TP and AZn contents were significantly and positively correlated with the bacterial genera norank_f__norank_o___norank_c__bacteriap25 and MND1 and significantly and negatively correlated with Rubrobacter, Ellin6055, and unclassified_f__Micromonosporaceae. OM, TN, AP, EMg, AFe, ACu, and HN were significantly and positively correlated with the bacterial genus Marmoricola and significantly and negatively correlated with norank__f__Gemmatimonadaceae, Solirubrobacte, and Gaiella. ECa and pH were highly significantly positively correlated with the bacterial genera norank__f__Gemmatimonadaceae, Solirubrobacter, and Gaiella and highly significantly negatively correlated with Marmoricola. The TK content was highly significantly positively correlated with the bacterial genera Arthrobacter and Sphingomonas and highly significantly negatively correlated with norank__f__norank__o___norank__c__MB-A2-108 and unclassified__k__norank__d__Bacteria. There was a highly significant positive correlation between the AK content and the bacterial genera Rubrobacter and Ellin6055 and a highly significant negative correlation with MND1.

Figure 11 Correlation analysis between the level of dominant species of soil microorganisms and the physicochemical parameters of the soil of Scutellaria baicalensis.

(A) Bacterium, (B) fungus. The presence of an asterisk indicates the presence of variability. p < 0.05 indicates that the index is significantly different between groups; * p < 0.05, ** p < 0.01, *** p < 0.001.

Figure 11B shows that the TP and AZn contents were significantly positively correlated with the fungal genera Solicoccozyma and Titaea and with Boeremia and unclassified_f__Arthopyreniaceae. In addition, TP was significantly positively correlated with the fungal genus Mortierella. The HN content was highly significantly positively correlated with the fungal genus Talaromyces and significantly negatively correlated with Metarhizium. OM, TN, AP, EMg, AFe, and ACu contents were significantly positively correlated with the fungal genera Tausonia, Talaromyces, Minimedusa, and Phoma and significantly negatively correlated with Metarhizium. ECa and PH were significantly and positively correlated with the fungal genus Metarhizium and significantly and negatively correlated with Tausonia and unclassified_f__Arthopyreniaceae. The TK content had a highly significant positive correlation with Cercospora, Botrytis, and Alternaria and a highly significant negative correlation with unclassified__k__Fungi and Isaria. The AK content was highly significantly positively correlated with Boeremia and unclassified__f__Phaeosphaeriaceae and highly significantly negatively correlated with Titaea and Solicoccozyma.

Discussion

Analysis of the active compounds of Scutellaria baicalensis cultivated from different sources in the Chengde area

The Scutellaria from Chengde, Hebei Province, is recognized for its superior quality and efficacy (Liu et al., 2024a). Jiang (2018) believes Chengde to be both the centre of origin and diversification for Scutellaria baicalensis. Research on Scutellaria baicalensis from this region holds significant value. The 2020 edition of the Chinese Pharmacopoeia stipulates baicalin as the quality control index of Scutellaria baicalensis herbs, and its content should not be less than 9.0%. The quality of the Scutellaria baicalensis herbs collected in this study was tested to meet the standard. The simultaneous detection of baicalin, baicalein, oroxindin, wogonin, and oroxylin A in Scutellaria baicalensis herbs via quantitative analysis of multiple components via the single-marker (QAMS) method revealed that there was variability in the active compounds of Scutellaria baicalensis from different places of origin. Among the three samples, PQ had the highest content of baicalin, followed by FN, and KC had the lowest content of baicalin. PQ had the highest contents of oroxindin and wogonin, and FN had the lowest contents of oroxindin and wogonin. KC had the highest content of baicalein and oroxylin A, and FN had the lowest content of baicalein and oroxylin A. Modern pharmacological studies have shown that baicalein and oroxindin have antitumour effects (Pan et al., 2016). Baicalin has antibacterial (Zhang et al., 2020), anti-inflammatory (Guo et al., 2013) and antioxidant (Wen et al., 2013) effects. On the basis of the variability of different active compounds of Scutellaria baicalensis from different origins, it can be used in clinical preparations to treat corresponding diseases according to different active compounds from different origins. The next steps should be to strengthen the study of the geographical adaptability characteristics of Scutellaria baicalensis herbs, rationally select planting locations, optimize the planting technology, and establish a scientific and reasonable quality evaluation standard for local herbs. Considering the small sample collection volume, the collection volume should be increased in future sample testing to enrich and further validate the experimental results.

Relationships between the active compounds of Scutellaria baicalensis and soil physiochemistry

Soil pH, heavy metal elements, trace elements, and quick-acting nutrients can affect the growth and development of medicinal plants and even the content of active compounds (Wu et al., 2013). The results of the correlation analysis revealed that the baicalein and oroxylin A contents were highly significantly positively correlated with the total phosphorus, effective manganese, and effective zinc contents, whereas the wogonin content was highly significantly negatively correlated with the total phosphorus, effective manganese, and effective zinc contents. The same elements had different effects on the accumulation of the active compounds of Scutellaria baicalensis. There was a negative correlation between baicalin content and organic matter, which was consistent with the results of Jiang (2021). Sun et al. (2020) reported that the abundance of Fe and P in the soil is favourable for the growth of high-quality Scutellaria baicalensis. In this study, we showed that the contents of baicalein and oroxylin A were highly significantly positively correlated with the effective Fe and effective phosphorus contents, whereas baicalin and wogonin were negatively correlated with the effective Fe and effective phosphorus contents. Nitrogen, phosphorus, potassium, calcium, magnesium, sulfur, iron, manganese, copper, and zinc affect the synthesis and metabolism of flavonoids in medicinal plants through the regulation of carbon and nitrogen metabolism, the metabolism of endogenous hormones in plants, and the activity of key enzymes (Liu et al., 2010).

Relationships between rhizosphere soil microorganisms and the active compounds of Scutellaria baicalensis and soil physiochemistry

Microorganisms or hosts that interact with microorganisms can provide a variety of bioactive components that are the primary chemical ingredients of Scutellaria baicalensis (Bulgarelli et al., 2012). Through long-term adaptation to the growth environment, the internal environment of medicinal plants controls the accumulation of secondary metabolites (Falcone Ferreyra, Rius & Casati, 2012; Guo et al., 2013). Certain plant functions, such as plant development and the accumulation of active chemicals, are even entirely dependent on the activity of microbes (Xie, 2016). Analysing the correlations among the physicochemical properties, soil microbes, and active compounds of cultivated Scutellaria baicalensis soil is useful for ensuring the quality of Scutellaria baicalensis medicinal herbs. Correlation analysis revealed that the bacterial genera Rubrobacter, Ellin6055, Gaiella, norank__f__norank__o__norank__c__bacteriap25, unclassified__f__Micromonosporaceae, norank__f__ Gemmatimonadaceae,

Arthrobacter, and Sphingomonas and the fungal genera Tausonia, Minimedusa, Cercospora, Botrytis, Alternaria, Boeremia, Titaea, Solicoccozyma, and Mortierell were both positively or negatively correlated with individual active compounds of Scutellaria baicalensis and were correlated with soil physicochemistry to varying degrees. Moreover, these genera are the dominant genera in terms of species composition, and the quality of cultivated Scutellaria baicalensis can be further ensured by clarifying the functions of these genera, which act as biotrophic bacteria or inhibit antagonistic bacteria.

Arthrobacter spp. can fix nitrogen, solubilize phosphate, detoxify potassium, and create indole-3-acetic acid (IAA) when they colonize plant roots. These processes greatly improve the fresh weight of plants (Jiang et al., 2022). In addition, Arthrobacter spp. have the ability to create a fibre matrix, which works against some plant pathogens (Chhetri et al., 2022). Sphingomonas spp. have been shown to promote plant growth and increase plant stress tolerance (Asaf et al., 2020). Interestingly, certain Sphingomonas spp. can interact with plants and function as culturable endophytic and rhizosphere bacteria. Alternaria spp. are well-known invasive pathogens and soil-borne pathogens (Liu et al., 2023). More than 95% of Alternaria spp. can colonize plants and cause plant diseases. Black spot disease caused by Alternaria spp. can cause plant diseases in medicinal plants such as Ginseng and Atractylodes lancea, affecting the yield and quality of medicinal plants (Schmey et al., 2024). As a soil phosphorus-solubilizing fungus, Mortierella spp. are able to promote the uptake of phosphorus by plants and promote plant growth. The mechanism is presumed to be that Mortierella spp. are able to secrete organic acids and other substances to directly activate insoluble inorganic phosphorus and organic phosphorus and can also indirectly activate phosphorus by interacting with the arbuscular mycorrhizae (AM) to promote phosphorus uptake and utilization by plants, thus promoting plant growth (Qiu et al., 2024). The results of this study also confirmed the significant positive effect of Mortierella spp. on the accumulation of total phosphorus. In the soil of cultivated Scutellaria baicalensis, the fungal genus Mortierella spp. and the bacterial genera Alternaria spp., and Sphingomonas spp. are all prophylactic bacteria that can further increase their abundance and promote the accumulation of effective compounds in Scutellaria baicalensis; they can also further increase the content of all-potassium and all-phosphorus compounds, increasing the content of active compounds of Scutellaria baicalensis by increasing the content of soil physicochemical factors. Alternaria spp. are pathogenic bacteria that can cause plant diseases; they can be used as plant-induced resistance agents to activate the plant’s own immune system to cope with disease and improve the quality of the plant. The metabolism of rhizosphere microorganisms affected the secondary metabolites of Scutellaria baicalensis in two main ways: on the one hand, it promoted the synthesis and accumulation of secondary metabolites of Scutellaria baicalensis. Specific bacteria and fungi were significantly correlated with the accumulation of secondary metabolites in Scutellaria baicalensis (Li, 2022). On the other hand, rhizosphere microorganisms influence metabolic processes, and certain microorganisms induce the expression of specific genes in Scutellaria baicalensis, activate specific secondary metabolic processes, and produce active chemicals (Ouyang et al., 2023). Overall, these are important for improving the quality and yield of Scutellaria baicalensis herbs.

Bacterial and fungal function prediction

To determine the functional prediction information of the bacteria in various samples, functional analysis of the bacterial colonies was carried out via PICRUSt2 software. All of the samples involved six major categories of biometabolic pathways, including metabolism, genetic information processing, environmental information processing, cellular processes, human diseases and organismal systems, according to a comparison of the sequencing data via the Kyoto Encyclopedia of Genes and Genomes (KEGG) database. These include metabolic processes, the processing of genetic and environmental information, cellular functions, human illnesses, and organismal systems. With metabolic processes enriching the majority of pathways, metabolism serves as the primary function. One of the main roles of metabolism in bacterial communities is to support plant growth and the cycling of soil materials (Moreno et al., 2019). A total of 46 subfunctions, including those related to immune illness, carbohydrate metabolism, transcription, and amino acid metabolism, were identified through additional examination of the predicted genes at the secondary functional level. The majority of the secondary functions are associated with the metabolism of amino acids, carbohydrates, and global and overview maps (Fig. 12). As the primary energy sources for soil microorganisms and the components or metabolites of microbial cells, amino acids and carbohydrates rank among the most significant functional genes in the bacterial community (He et al., 2020). Soil microorganisms, such as Streptomyces and Bacillus, promote plant growth and crop yield through metabolic activities through metabolism to produce antibiotics, bacteriostatic proteins and plant growth hormones to achieve the control of pathogenic bacteria and promote plant growth, nitrogen fixation, phosphorus solubilization, etc., through metabolism to promote the absorption of nitrogen and phosphorus nutrients in the root system (Wakelin et al., 2017). Carbohydrate metabolism is closely related to nitrogen fixation and phosphorus solubilization, which help promote nitrogen and phosphorus cycling in plants, e.g., Arthrobacter and Sphingomonas (Bromke, 2013).

Figure 12 Statistical heatmap of KEGG functional abundance.

Fungal functional groups are primarily related to the degree of fungal dependence on the plant host. Fungal function prediction via FUNGuild revealed that unknown and undefined saprotrophs (undefined saprotrophs), endophyte-litter saprotroph-soil saprotroph-undefined saprotrophs (endophyte-litter saprotroph-soil saprotroph-undefined saprotroph), and animal pathogens were present in greater proportions in the three sample plots (Fig. 13). The proportion of animal pathogen ecological function in FN was 26.96%, which was significantly greater than that in KC and PQ (0.92% and 1.13%, respectively), indicating that the main fungal function in FN was animal pathogenesis. The relative abundances of the ecological functions of the endophyte-apomorphytic humus-soil humus-undefined humus category were 12.05%, 10.01%, and 3.02% in KC, FN, and PQ, respectively. The functions of the dominant fungal communities in the rhizosphere soil of Scutellaria baicalensis cultivated from different origins differed significantly. Saprophytic fungi are associated primarily with plant growth and metabolic activity and are able to provide additional nutrients to plants by degrading apoplastic material and humus (van der Heijden, Bardgett & van Straalen, 2008). Pathogenic fungi are associated with plant community diversity, primarily by infesting specific hosts and altering patterns of interspecific competition in vegetation (Bever, Mangan & Alexander, 2015). Changes in the environment and nutrient inputs due to anthropogenic disturbances to the soil and fertilization may lead to functional switching of the fungal community by altering the environment in which the fungi live.

Figure 13 FUNGuild functional classification statistics bar chart.

Conclusion

The aim of this study was to analyze the differences in the composition, structure, and diversity of the soil microbial community between the roots of cultivated Scutellaria baicalensis as well as the correlation between soil microbes and the active compounds of Scutellaria baicalensis as well as the soil physicochemical factors, to further clarify the influence of soil microbes on the accumulation of the active compounds of Scutellaria baicalensis, and to provide theoretical support for the artificial cultivation of Scutellaria baicalensis and standardization of cultivation bases. There was variability in the composition of the rhizosphere soil microbial community and in the contents of the active compounds of Scutellaria baicalensis from different origins. The diversity of the soil microbial community structure in cultivated Scutellaria baicalensis was correlated with soil physicochemical factors and the content of active compounds of Scutellaria baicalensis, which, to a certain extent, indicated that the composition of the soil microbial community and soil physicochemical factors affected the synthesis of active compounds of Scutellaria baicalensis. Important genera that were closely related to soil physicochemistry and the active ingredients of Scutellaria baicalensis include the fungal genera Tausonia, Minimedusa, Cercospora, Botrytis, Alternaria, Boeremia, Titaea, Solicoccozyma, and Mortierella. The bacterial genera Rubrobacter, Ellin6055, Gaiella, norank__f__norank__o_norank__c__bacteriap25, Arthrobacter, unclassified__f__Micromonosporaceae, norank__f__Gemmatimonadaceae, and Sphingomonas were identified. These findings provide critical information for the commercial production/cultivation of high-quality Scutellaria baicalensis for medicinal purposes. A selected group of growth-promoting microorganisms, as listed above, can be harnessed as biofertilizers to increase the quality of medicinal products, not only for Scutellaria baicalensis but also for other similar plants.

Supplemental Information

Supplemental Information 1 Content of active compounds of Scutellaria baicalensis..

Supplemental Information 2 Physicochemistry of the rhizosphere soil of cultivated Scutellaria baicalensis..

Supplemental Information 3 Relative abundance of dominant bacterial genera.

Supplemental Information 4 Relative abundance of dominant fungal genera.

Additional Information and Declarations

Competing Interests

Author Contributions

Data Availability

The authors declare that they have no competing interests.

Ping Dong conceived and designed the experiments, performed the experiments, analyzed the data, prepared figures and/or tables, and approved the final draft.

Yang Lu conceived and designed the experiments, authored or reviewed drafts of the article, and approved the final draft.

Jia Bai performed the experiments, authored or reviewed drafts of the article, and approved the final draft.

Chunying Zhao analyzed the data, authored or reviewed drafts of the article, and approved the final draft.

Zhongsi Li analyzed the data, authored or reviewed drafts of the article, and approved the final draft.

Yu Cao analyzed the data, authored or reviewed drafts of the article, and approved the final draft.

Yingzhe Li performed the experiments, authored or reviewed drafts of the article, and approved the final draft.

The following information was supplied regarding data availability:

The data is available at NCBI: SRP531568.

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
