# Peer review of "The influence of rhizosphere microbial diversity on the accumulation of active compounds in farmed Scutellaria baicalensis"

_PeerJ, doi:10.7717/peerj.18749_

## Round 0.1 · original submission · Major Revisions

Recommendations of the three reviewers range from 'Minor Revisions' to 'Reject'. All three reviewers highlight major issues in the manuscript. Among many other issues, Reviewers 1 and 2 highlight the discrepancy in the correlation analysis. Reviewer 3 provides extensive comments on the taxonomic nomenclature.

Apart from the major issues on the content of the manuscript, all reviewers highlight issues on the clarity of presentation and language usage. In particular, all three reviewers state that the presentation is confusing.

The authors are requested to address ALL comments of all reviewers with a point-by-point explanation of how each comment has been addressed.

·

Basic reporting

This article analyzes the relationship between soil physicochemical properties and microbial composition of Scutellaria baicalensis rhizosphere. This type of work is essential to understanding how the rhizosphere microbiome affects the plant's medicinal properties. From this perspective, the given article is valuable. Their experimental setup is sound and they have conducted a good statistical analysis. However, the writing and formatting of the article is very poor. The grammar and the English writing have to be improved significantly. And the formatting of the figures and the article should be significantly enhanced. Several figures are not referenced in the text and certain statistical tests lack consistency. For instance, in the methods, they say they used Pearson’s correlation but this has changed to Spearman’s correlation in the results. Without these improvements, the article should not be published. My specific detailed comments are given below.

Abstract and Introduction
Throughout the article, the scientific names are not italicized. This should be fixed ( e.g, lines 38,39).
Some grammar mistakes throughout (e.g., lines 42, 71, and many more!)
No spaces after the period (e.g., lines 43, 45, and many more!)
Citations are needed for some places (e.g., line 52)
Line 52: Actinomycetes is also a bacteria

Experimental design

The research gap and the research goals are not properly explained in the introduction. This should be added to the last part of the introduction.

Specific comments for Materials and Methods
Section 1.2.2: If possible, it is better to add citations for each method.

In section 1.3, line 136, you mention that the Pearson correlation was used to elucidate the relationships between the soil conditions and the 5 bio-compounds. However, Pearson correlation should be used if the data is normally distributed. Have you performed any normality tests? Otherwise, you have to use alternatives such as Spearman correlation.

Validity of the findings

Line 144: The Figure 1 caption does not properly explain the figure.
Line 173: Spacing issues and a period are used instead of a comma.
Again, the Figure 2 caption is incomplete. Horizontal ven diagram of what?
Line 192: The same thing is repeated.
Line 201: Why do you talk about species? your bar graphs are only for genera. And I do not understand what you meant by combining with other species?
Line 204: Table 1 is from supplementary files. This is not clearly indicated. And it only has genera, not species.
Figure 6: How were these 4 genera selected for the comparison?
Line 219: From what figure?
Line 235-236 The English writing is not good. Reword.
Line 249: Here you say, you used Spearman correlation, but in the methods, you say you used Pearson correlation. Which one is it?
Line 252: Why do you have raw names like “norank_f__Gemmatimonadaceae” instead of properly naming the genera?
Line 255: Grammar errors
Section 2.5: No figure is referred to when explaining the results. Figure 9 should be referenced.
Section 2.6: Again, no figure was referenced.

Additional comments

In general, the discussion lacks some significant information. For instance, the alpha and beta diversities are not properly discussed.
Line 326: Grammar mistakes: create a fiber matrix
The last two figures (Figures 11 and 12) are not referenced in the text.

Reviewer 2 ·

Basic reporting

The author intend to reveal the infuence of microbial diversity on the accumulation of active substances in the inter-root soil of farmed Scutellaria baicalensis. There are some comments:
1.The paper still needs to be doctored into a better version. The logic in the introduction is weak.
2.What is Inner-root mean? root endophyte? Or rhizosphere?
3.Line 29-30, “norank__f__norank__o_norank__c__bacteriap25” should be revised. You can't just copy it from the sequencing company's report.
4.Line 117-138, References should be given to the software used in the methods section.
5.Line249, the author describes that “Using Spearman's correlation heatmap to analyze the relationships between the effective”. But in the method section, the author says that “using Pearson's straight-line correlation analysis”. So which method did you use?
6.Line 350-375, I CAN’T find the results of functional analysis of the bacteria. But in the discussion section, the author discusses the functional analysis.
7.The bacterial genera and fungal genera should be italic.
8. where is the raw data?

Experimental design

1. The method of "1.1 Sample collection and processing" was to simple. Readers can't get detail information from this section.

Validity of the findings

1. Correlation analysis was performed in the study. While Fewer sample may cause huge bias. More samples should be collected if a correlation analysis was used.

Additional comments

no comment

Reviewer 3 ·

Basic reporting

The title of the research article is of critical importance to contemporary and sustainable agricultural production. It adequately captures the article’s content in the sense that rhizosphere microbial diversity of Scutellaria baicalensis coupled with soil physicochemical properties has an influence on the medicinal active compounds produced by the plant. This subject is relevant and interesting as well. I would have preferred the title to read: "The influence of inter-root soil microbial diversity on the accumulation of active substances in farmed Scutellaria baicalensis". Overall, the manuscript is well written.

Experimental design

The experimental designs of the manuscript are well thought out, appropriate and quite intense. Their outcome as demonstrated in this manuscript address the challenges at hand.

Validity of the findings

The findings of the manuscript are valid though with reservations in certain aspects as outlined below:
 Page 4 lines 142-144; The authors need make statistical comparisons for each of the three active compounds separately and not use one sentence as done here.
Authors need to report the differences for all the detected active compounds in the three sampling areas. The reporting done in the current manner is not clear though it is reflected in figure 1.
 Page 5 line 189; The text provided in this line is quite confusing based on figures 3 and 4. First, why was a Kruskalis-Wallis rank sum test taken? What was it supposed to achieve? What hypothesis originated from its testing? What were the variables (both dependent and independent)? I can't see the outcome of this testing anywhere.
Alpha diversity analysis usually involves comparing richness, evenness and diversity in a community. The most common analysis expected includes Simpson's indices, Chao 1 indices and Shannon's indices. This study has adopted the last two which suffices. The Kruskalis-Wallis rank sum test is not even an alpha diversity analysis. The authors need to get rid of it.
 Page 5 lines 189-190; The reported results are NOT what is depicted in Figure 3A! There is no significant difference in the Shannon indices of the three samples.
 Page 5 line 191; Reported results are NOT true as seen on figure 3
 Page 5 line 192; The text reporting has not been shown on the error bars of Figure 3B. The authors need to give statistical differences using lower case alphabetical letters, preferably a, b and c.
 Page 5 lines 193-194; Based on Figure 4 A and B, what has been reported in the text is NOT true.
 Page 5 line 195; The authors intimate that the fungi sampled were endophytic. That is not the case because they were not isolated from the interior of the plant. They were rather rhizospheric.
 Page 5 line 195; The authors intimate have indicated that the environments presented in the three samples of Scutellaria baicalensis were the same. That is not true.
 Page 5 lines 195-197; The authors intimate that the diversity of bacterial communities in the inter-root soil of the species from the three sample sites was highly significant. Again, that is not accurate based on the results presented in figures 3 & 4
 The entire Discussion: Page 8 lines 303-375; The authors need to be sensitive to basic established scientific rules. One such rule is the writing of scientific names (Bionomial Nomenclature). The first letter of the genus name is always capitalized/upper case and the specific epithet has the lower case. The two names are italicized. This should always be observed. This also applies to the entire article
 Page 9 lines 331-332; Streptomyces is a bacterium in the phylum Actinomycetota and the type species of the family Streptomycetaceae. It is one of the soil bacterium that is known to produce a considerable number of antibiotics (two-thirds of the clinically useful antibiotics). Alternaria on the other hand is a fungus in the Division Ascomycota, family Pleosporaceae. The two are very different entities. The authors, in the text provided, seem to indicate that the two are the same. This is not the case and they need to correct this and consequently amend the discussion they are presenting in this article about these two.
 Page 9 line 340; Aspergillus was not among the fungal genera that the study identified to be associated with Scutellaria baicalensis rhizosphere soil according to figure 5B. I am not sure why the authors are discussing it.
 Page 9 line 347; Streptosporium spp. was not isolated from plant-inter root soil and therefore should not be discussed.
 Pages 9 & 10 lines 350-375; There is very little discussion of the role of metabolism of the plant inter-root fauna in relation to their influence in the production of secondary metabolites in Scutellaria baicalensis. It is one of the most critical points of discussion that should add value to this article. A succinct discussion is required.

Additional comments

I have the following comments on the other sections of the manuscript:
1. Introduction
 Page 1 line 38; I am not sure which species Xenospora baicalensis Georgi is! Reading it in this article made me totally confused! Xenospora is a genus of moths in the family Geometridae, order Lepidoptera. How this genus name finds its way to become a plant species is a mystery. Following this statement, the authors seem to suggest that the dried root of Scutellaria baicalensis Georgi. is referred to as Xenospora baicalensis. This is not the case. If anything, the plant has six synonyms that came about with the dynamics of systematics, viz:
i) Scutellaria macrantha Fisch. ex Rchb. In Iconogr. Bot. Pl. Crit. 5: 52 (1827)
ii) Scutellaria adamsii A.Ham. In Esq. Monogr. Scutellaria: 34 (1832)
iii) Scutellaria davurica Pall. ex Ledeb. In Fl. Ross. 3: 397 (1849)
iv) Scutellaria speciosa Fisch. ex Turcz. In Bull. Soc. Imp. Naturalistes Moscou 24(II): 389 (1851)
v) Scutellaria lanceolaria Miq. In Ann. Mus. Bot. Lugduno-Batavi 2: 110 (1865)
vi) Scutellaria baicalensis f. albiflora H.W.Jen & Y.J.Chang In J. Beijing Forest. Univ. 13(3): 3(1991)

However, the roots of Scutellaria baicalensis Georgi are referred to as Radix Scutellariae or Huang Qin in Chinese, the primary source of its medicinal properties (Pei et al., 2022); http://doi.org/10.1186/s12864-022-08391-1

 Page 2 lines 66-72; This sentence in these lines is too long and I cannot clearly get what the authors are communicating. I suggest it be divided into at least three sentences to properly articulate the message with regard to: the synergistic relationship between Scutellaria baicalensis and inter-root microbes and their effect on the quality of medicine from the plant, the health status of the plant herbs and provision of a foundation to screen plants of excellent quality.

2. Materials and Methods
 Page 2 line 76; I suggest the authors provide a figure showing a map of the sampled areas/counties around Chengde City within Hebei Province. If possible, indicate the points on the map where sampling was done. In this day and era, I guess geographical coordinates were taken to make it easy to reproduce them on a map. This will add a lot of value to the methodology.
 Page 2 line 79; While sampling there is an element of taking samples from different points and getting composite ones. In this case, the authors are alluding to 3. Why not the 5 given the use of the 5-point sampling method?
Secondly, it will be useful for the authors to indicate the approximate amount of root soil from each sample and what amount constituted the composite sample for purposes of replicating this experiment
 Page 2 lines 77-81; The authors need to expressly give references of the methods adopted. I strongly recommend a brief outline of the methods used to give the audience an idea of what exactly was done while reading the article rather than just being referred to a reference.
 Page 3 line 84: subtitle 1.2; I recommend replacement of "effective components" with "active compounds" throughout the article.
 Page 3 lines 89-98: subtitle 1.3; The authors need to give references for the methods mentioned in this section. Thus, insert references for: i) The Kjeldahl method, The Molybdenum antimony calorimetric method, etc.

3. Conclusions
 Page 10 lines 378-380; I am not convinced that the text provided in these lines is the the purpose of the study as alluded to by the authors. Reading this article makes me agree with the title of the article that the authors were seeking to establish the influence of inter-root soil microbial diversity on the accumulation of active compounds by Scutellaria baicalensis achieved by analysis the relationships between physicochemical properties of soil, the diversity of microorganisms and influence on their metabolic pathways in the production of active compounds by the plant. The text needs to be revisited and corrected.

4. Figures and their captions
i) Figure 3; The Figure needs a proper caption that explains the statistics
ii) Figure 4: The authors need to show statistical differences on the figure using lower case letters. It can be misleading if one relied on visual observation alone.

5. Tables and their titles
i). Supplemental Table 1; Statistical analysis is missing in this table. This should be inserted
ii). Supplemental Table 2; Statistical analysis is missing in this table. This should be inserted. That begs the question; How did the authors arrive at their conclusions?

Annotated reviews are not available for download in order to protect the identity of reviewers who chose to remain anonymous.

---

## Round 0.2 · accepted · Accept

The reviewers are satisfied that the authors have addressed all issues raised by them.

Reviewer 2 ·

Basic reporting

The author has made detailed revisions based on the previous round of revision comments.

Experimental design

The author has made detailed revisions based on the previous round of revision comments.

Validity of the findings

The author has made detailed revisions based on the previous round of revision comments.

Reviewer 3 ·

Basic reporting

I have re-reviewed the manuscript and it now makes sense. The grey areas have been addressed. It is generally well done and meets the publication criteria of PeerJ.

Experimental design

I'll say again that the experimental designs of the manuscript are well thought out, appropriate and quite intense. Their outcome as demonstrated in this manuscript address the challenges at hand.

Validity of the findings

The findings of the manuscript are certainly valid and of great import to the relevant fields: Medicine, Commercial Agriculture

Additional comments

I have no additional comments

Annotated reviews are not available for download in order to protect the identity of reviewers who chose to remain anonymous.